# Chemotaxis of *Meloidogyne incognita* Response to Rhizosphere Bacteria

**DOI:** 10.3390/microorganisms11092271

**Published:** 2023-09-09

**Authors:** Beiyang Li, Pinyi Wang, Liangliang Yang, Xiaozhan Rang, Wenzhen Zhou, Yajun Liu

**Affiliations:** State Laboratory for Conservation and Utilization of Bio-Resources, Yunnan University, Kunming 650091, China; libeiyang@itc.ynu.edu.cn (B.L.); 15587113964@163.com (P.W.); 18987684480@163.com (L.Y.); rangxiaozhan@itc.ynu.edu.cn (X.R.); 12021114082@mail.ynu.edu.cn (W.Z.)

**Keywords:** rhizosphere bacteria, *Meloidogyne incognita*, chemotaxis, volatile organic compounds

## Abstract

Rhizosphere microorganisms and the volatile organic compounds (VOCs) produced by them take part in the regulation of the chemotaxis of nematodes. A total of 150 strains of rhizosphere bacteria were screened via a chemotaxis experiment with *Meloidogyne incognita*. Some isolates affected the behavior of the nematodes, including attraction, randomness, and repulsion. Volatile metabolites produced via the selected bacteria were associated with the chemotaxis of nematodes. *M. incognita* was highly attracted to decanal. In addition, dimethyl disulfide, 2,5-dimethylpyrazine, pentadecanoic acid, and palmitic acid were found to attract weakly *M. incognita*. Furthermore, the chemotaxis of *M. incognita* was tested in a pot experiment. The bacteria *Bacillus* sp. 1-50, *Brevibacillus brevis* 2-35, *B. cereus* 5-14, *Chryseobacterium indologens* 6-4, and VOC decanal could regulate the movement of M. incognita in the pot with or without plants. The results provide insights into rhizosphere microorganisms and their VOCs and how they regulate the chemotaxis of the nematodes.

## 1. Introduction

Nematodes use a head-sensing organ (amphid) to perceive external environmental stimuli, and they tend to move toward beneficial stimuli and away from harmful stimuli [1]. We define these behaviors as nematode chemotaxis. Therefore, it is important for the survival of nematodes, such as in finding food or a host, avoiding pathogenic bacteria, and mating [2,3].

*Caenorhabditis elegans* recognizes and locates signals using an olfactory organ [4,5]. Moreover, they can be acutely aware of pathogens and move away from them [6,7]. For plant-parasitic nematodes (PPNs), chemotaxis is important to recognize host plants [8,9]. Previous reports have shown that the exudates of the host root influence the chemotaxis behaviors of PPNs, such as attraction and repulsion [10,11,12,13,14]. In addition to host roots, soil microbes also influence the behavior of nematodes. Cheng et al. [15] isolated 11 volatile organic compounds (VOCs) from strain *Paenibacillus polymyxa* KM2501-1, and five compounds had chemotactic activity for *M. incognita* (acetone, 2-decanol, furfural acetone, 2-undecanone and 4-acetylbenzoic acid). Hao et al. [16] analyzed soil bacteria that had attractive activities to *Panagrellus redivivus* and found 11 volatiles produced via bacteria with nematode-attracting activity. *Heterorhabditis bacteriophora* was attracted via 1-heptanol, 1-octanol, and 1-nonanol, which are byproducts of bacterial metabolism [17].

The chemotaxis of nematodes is closely linked to the rhizosphere microorganisms [7,18]. *Tylenchorhynchus ventralis* prefers roots that lack their microbial enemies [19]. *M. incognita* penetration is reduced in tomato roots colonized *arbuscular mycorrhizal* fungi [20]. The VOCs produced via soil microorganisms would have important functions in determining the behavior of nematodes. The chemotaxis activities of soil bacteria to *M. incognita* were screened in this study. Volatile compounds from microorganisms were identified using GC-MS. The influence of microbes and VOCs on the nematode chemotaxis behavior was evaluated via Petri dish and pot experiments to improve our understanding of the interactions between soil microorganisms and nematodes.

## 2. Materials and Methods

### 2.1. Bacteria Strains and Nematodes

Strains of bacteria were isolated from the rhizosphere of tomatoes in Ershan County (101°52′–102°37′ E, 24°01′–24°32′ N), Yunnan Province, China, and separated via the dilute coating plate method [21]. The rhizosphere of soil suspension was successively diluted by 10 times gradient with sterile water and coated on 9 cm Petri dishes containing Luria-Bertani (LB) medium (37 °C, 16 h). Then, the single colony was selected from the LB medium. The isolated strains were maintained in 10% glycerol (−80 °C). When tested, the bacteria were activated by shaking in LB liquid medium (37 °C, 180 rpm).

The collection and maintenance of *M. incognita* followed the methods by Wang and Lu et al. [22,23]. After 6 weeks of infection, the egg masses were separated using manual pick from the tomato root. Egg masses were disinfected with 0.1% sodium hypochlorite (1 min) and washed five times with sterile water in an Eppendorf tube in a Vortex shaker (200× *g*, 1 min). Juveniles (J2s) were collected onto a 50 μm mesh sieve in distilled water (25–30 °C).

### 2.2. Chemotaxis Assay

Chemotaxis was tested using a modified version of the method from Wang et al. [23,24]. The chemotaxis assays were performed at 28 °C in the dark. The chemotactic index (C.I.) in the Petri dish was calculated after 4 h. According to previous reports [24]. The C.I. of each strain was calculated from three plates (counting nematodes under a microscope) and replicated three times. Similarly, the sample for the VOCs was subjected to chemotaxis experiments, as shown above. The VOCs were dissolved in ethanol, and the control solution was ethanol.

### 2.3. Identification of Bacteria Strains by Ribosomal DNA Sequencing

DNA from the isolates was extracted using the lysozyme method with some modifications. Then, 50 mg of isolates were collected via centrifugation at 1000× *g* for 1 min. The pellet was resuspended in 480 μL 1 × TE (100 mL 1 M Tris-HCl, 20 mL 0.5 M EDTA, pH = 8.0) and 20 μL lysozyme solution (50 mg/mL) for 10 h at 37 °C. Then, 50 μL 20% SDS and 5 μL 20 μg/mL Proteinase K were added, shaken for 1 min, and incubated for 30 min at 55 °C. A total of 550 μL phenol/chloroform/isoamyl alcohol (25:24:1) was added and vortexed to mix thoroughly. Centrifugation was performed at 6000× *g* for 10 min, and the supernatant was absorbed. Crude DNA was recovered from the supernatant by adding 800 µL anhydrous ethanol and 80 µL 3 mol/L sodium acetate for 10 min at room temperature. After centrifugation at 6000× *g* for 10 min, the supernatant was removed, and 1 mL of 70% ethanol was added to wash the DNA twice (6000× *g*, 5 min, 4 °C). Finally, the DNA was resuspended in 500 μL 1 × TBE.

16S rDNA was extracted from the DNA samples using the universal eubacterial primers 8F and 1492R. The sequence (5′-3′) of the forward and reverse primers were as follows: AGAGTTTGATCCTGGCTCAG and GGTTACCTTGTTACGACTT.

Each 50 μL PCR reaction contained an 80 ng template, 2.5 mM MgCl 2, 2.5 units Taq polymerase (TaKaRa, Kusatsu, Japan), 0.24 mM each dNTP, 0.5 μM each DNA primer, and 1 × PCR buffer (TaKaRa). Reaction mixtures were held at 95 °C for 4 min, cycling 30 times in three steps: denaturation (98 °C, 10 s), annealing (56 °C, 45 s), and primer extension (72 °C, 1 min). The final step of the primer extension was carried out at 70 °C for 10 min. PCR fragments of 1500 bp were sequenced and compared using NCBI nucleotide blast stirring.

### 2.4. Analysis of Microbial VOCs by GC-MS

VOCs were collected using a SAAB-57318 (75 μm CAR/PDMS) SPME. A new SPME fiber was preconditioned with helium at 270 °C for 2 h before use. Extractions were performed in 15 mL Supelco SPME vials filled with 7 mL bacterial fermentation (OD = 1.0–1.2 A) containing a stirring rod. A thermostatic magnetic stirrer was placed on top of the vials. The SPME needle pierced the septum, and the fiber was exposed to the headspace of the vial for 120 min at 60 °C with constant magnetic stirring. Analysis of microbial VOCs was conducted via GC-MS.

VOCs produced via the representative isolates were detected using a 7890 GC/5975 MSD gas chromatograph/mass spectrometer (Agilent, Santa Clara, CA, USA) [23,24,25]. Identification of compounds was carried out via comparison of retention times and mass spectral data. Relative content was determined from the standard chart (spectrogram database NIST111L). Each data is the mean of three replicates.

### 2.5. Analyses of M. incognita Behavior in the Pot Experiment

The pot experiment was designed using the method described according to Wang et al. [23,24]. A polyvinyl chloride tube (Ø1 × 18 cm) was connected to two plastic cups (Ø 7 cm).

In assay I, sterilized irrigated sand soil (90 g), which was baked at 80 °C for 8 h, was placed in plastic cups and the polyvinyl chloride tube. Four bacterial broths (OD = 1.0–1.2 A) and the control (liquid LB medium) were added to the opposite plastic cup, respectively. The candidate VOC (decanal) and the control (10% aqueous ethanol) were also added to the opposite plastic cup, respectively.

In assay II, tomato seedlings (Jiabao, from Beijing Reach Modern Agriculture Development Company, Beijing, China) growing in a greenhouse at 25–30 °C for 20 days were transplanted into plastic cups. Then, four bacterial broths (OD = 1.0–1.2 A) and the control (liquid LB medium) were added to the opposite plastic cup, respectively. The candidate VOC (decanal) and the control (10% aqueous ethanol) were added to the opposite plastic cup, respectively.

Two thousand fresh J2s were injected into the hole in the middle of the polyvinyl chloride tube 10 days after adding the samples. About 24 h after adding the J2s, those in the polyvinyl chloride tube were collected. When soil samples were wrapped in lens paper and soaked in deionized water, nematodes crawled out of the soil, similar to Hoobermann tunnel method, and then, the number of J2 was calculated under electron microscope. Each process is repeated five times.

### 2.6. Statistical Analysis

SPSS 11.0 is used for the statistical analyses. The chemotactic index is shown as the mean ± standard deviation (SD) (*n* ≥ 3). Significant differences were assessed with a *t*-test (* *p* < 0.05; # *p* < 0.0001).

## 3. Results

### 3.1. Screening and Identification of Strains

Eventually, we isolated 150 strains of bacteria from the soil. All isolates were screened for the chemotaxis of *M. incognita*. The results showed that one isolate (0.6%) had highly attractive activity and seven isolates (4.67%) had highly repellent activities (Figure 1A). Finally, four isolates that attracted, repelled, or randomized to *M. incognita* in Petri dishes were selected for identification and further study (Figure 1B, Table 1).

### 3.2. The Chemotaxis of the VOCs Produced by Bacteria

Sixty-seven compounds were detected via GC-MS from the four isolated cultures, and the nine VOCs that were specific or common in the four isolates were selected for further testing (Figure 2; Table 2).

A total of nine VOCs were tested for the chemotaxis of *M. incognita* in Petri dish experiments (Table 3). *M. incognita* was highly attracted by decanal (1 mg/mL, C.I. = 0.20 ± 0.10) and weakly attracted by dimethyl disulfide (100 mg/mL, C.I. = 0.14 ± 0.02), 2,5-dimethyl pyrazine (10 mg/mL, C.I. = 0.16 ± 0.10), pentadecanoic acid (100 mg/mL, C.I. = 0.16 ± 0.02), and palmitic acid (100 mg/mL, C.I. = 0.11 ± 0.03).

### 3.3. M. incognita Behavior in the Pot Experiment

To evaluate the chemotactic behavior of *M. incognita* in the soil, we conducted a pot experiment. It showed that the J2s were attracted more to the four isolated cultures compared to the control at different doses in soil (Figure 3). The highest percent increases in the attraction over the controls (*p* < 0.05) were 267 ± 0.61%, 172 ± 1.29%, 108 ± 0.20%, and 545 ± 1.45% for *Bacillus* sp. 1-50, *B. brevis* 2-35, *B. cereus* 5-14, and *C. indologens* 6-4, respectively. As shown in Figure 3F, *M. incognita* moved randomly in the sterile soil on both sides (blank control). The attractive activities of the microorganisms were similar to those of tomatoes, which significantly attracted *M. incognita*, and the highest percent increase was 367.7 ± 0.04% compared to the control (sterile soil) (Figure 3E).

Moreover, we examined the responses of the nematodes to the isolated cultures in the presence of tomatoes (Figure 4). It showed that the J2s preferred the tomato without the addition of the cultures of *Bacillus* sp. 1-50, representing the highest decrease of 32 ± 0.16% relative to the control (*p* < 0.05), whereas J2s preferred the tomato with the other isolated cultures added, and the highest percent increases were 68 ± 0.03, 73 ± 0.30, and 114 ± 1.1 for *B. brevis* 2-35, *B. cereus* 5-14, and *C. indologens* 6-4, respectively, relative to the control (*p* < 0.05).

Meanwhile, we also explored the chemotaxis effects of volatile compounds (decanal) on J2s in the presence or absence of tomato plants. The results of the decanal showed that the J2s preferred decanal compared to the control (*p* < 0.0001), with the highest percent increase of 145 ± 0.27% (Figure 5A). In the preference for tomato, the J2s preferred the tomato with decanal relative to the control (*p* < 0.05), and the highest increase was 100 ± 0.34% (Figure 5B).

## 4. Discussion

The results suggested that the rhizosphere microorganisms affected the behavior of root-knot nematodes. As shown in the Petri dish assay, the cultures of *Bacillus* sp. 1–50 and *B. cereus* 5–14 attracted *M. incognita*. It is well known that *Bacillus* spp. is widespread in nature and has been extensively used for biological control [26,27,28]. *B. subtilis* and *B. thuringiensis* have been widely used as insecticides against nematodes [29,30,31]. Oka et al. [32] found that *B. cereus* inhibited the penetration of *M. javanica* second-stage juveniles into tomato roots in the soil. The control effect of *M. incognita* was up to 77.45%, with *B. cereus* BCJB01 after the addition of a nutrient matrix and organic fertilizer [33]. Huang et al. [34] found that *B. megaterium* YMF3.25 is an efficient biocontrol agent (BCA) against *M. incognita*. In our study, most of the attractive species were *Bacillus* spp., but there are few reports about the attractive activity of these bacteria for nematodes. The attraction will be a powerful boost to *Bacillus* spp. against plant parasitic nematodes.

It was first found that decanal significantly attracted *M. incognita*. Decanal has 100% nematicidal activity against *P. redivivus* and *Bursaphelenchus xylophilus* at a high concentration (6.7 mg/mL) [35]. Dong et al. [36] also found that lauric acid at low concentrations (0.5–2.0 mM) can attract *M. incognita* and consequently cause death, while high concentrations (4.0 mM) can repel it. Attractants can have different effects on nematodes at different concentrations. Furthermore, pentanoic acid and palmitic acid, detected in *B. cereus* 5-14, may be involved in the attraction of the strains, but their chemotactic activities are weaker than the strains, which suggests that the chemotaxis may be not induced only by a single compound.

The same attraction responses were observed in both the Petri dishes and the pot, such as for *Bacillus* sp. 1-50 and *B. cereus* 5-14. *M. incognita* was repelled via the cultures of *B. brevis* 2-35 and *C. indologens* 6-4 in the Petri dishes, but it was attracted to them in the pots. The microorganisms that grew on the Petri dishes may not act the same in the soil. Piśkiewicz et al. [19] found that *Tylorida ventralis* exhibits no significant difference in seeding with or without microorganisms in the Petri dishes, but it prefers seed without microorganisms in the Y-tubes with soil. At the same time, the J2s preferred the tomato with the strains of *B. brevis* 2-35, *B. cereus* 5-14, and *C. indologens* 6-4 compared to those without and were repelled by the tomato with *Bacillus* sp. 1-50, more so than those without, although that was highly attractive for *M. incognita* in the pot. The attraction of tomatoes was enhanced by the microbes, relative to the control. Furthermore, in contrast to microorganisms, the responses of the nematodes to the VOCs were consistent in the two systems. The stable chemotaxis activity of compounds is easier to develop than that of microorganisms. As the complex ecological environment in the soil is an obstacle to the detection, collection, and separation of allelochemicals, more signal compounds should be discovered in a further study.

Our results indicated that rhizosphere bacteria can regulate the chemical substances around plants and then influence the chemotaxis of root-knot nematodes. Chemotaxis further affects nematodes’ host location. This will help to understand how nematodes find their hosts. It also provides clues to prevent root-knot nematodes by blocking the nematode recognition of the host.

## Figures and Tables

**Figure 1 microorganisms-11-02271-f001:**
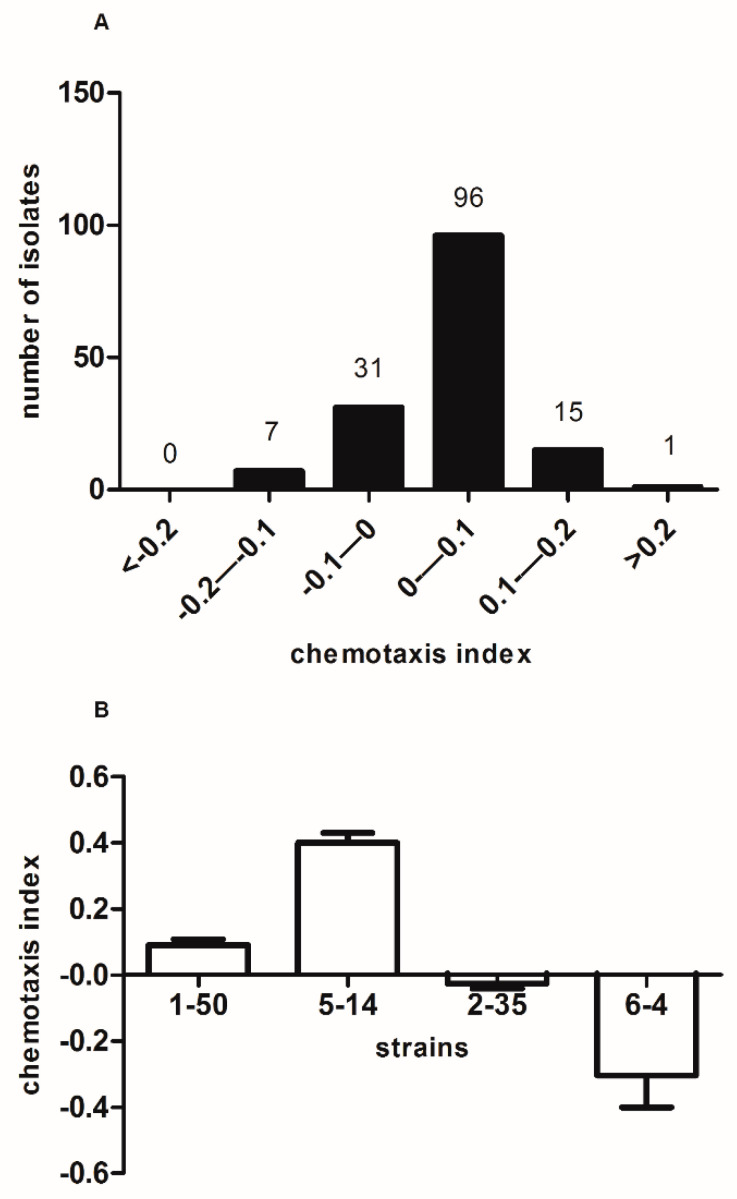
Chemotaxis index of *M. incognita* to microorganisms: (**A**) 150 tested bacteria and (**B**) 4 selected bacteria.

**Figure 2 microorganisms-11-02271-f002:**
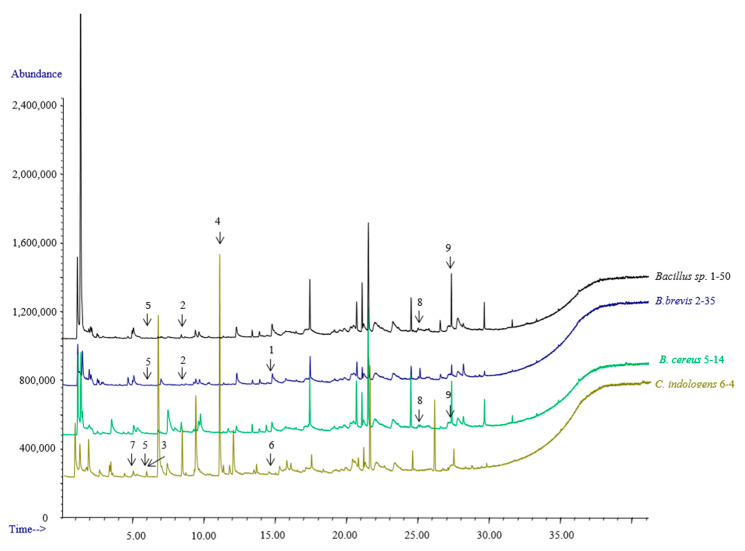
GC-MS analysis of the VOCs from selected strains. The arrows refer to the compounds: 1. Decanal, 2. Benzeneacetaldehyde, 3. Dimethyl disulfide, 4. 2-nonanone, 5. Benzaldehyde, 6. Indole, 7. 2,5-dimethyl pyrazine, 8. Pentadecanoic acid, 9. Palmitic acid.

**Figure 3 microorganisms-11-02271-f003:**
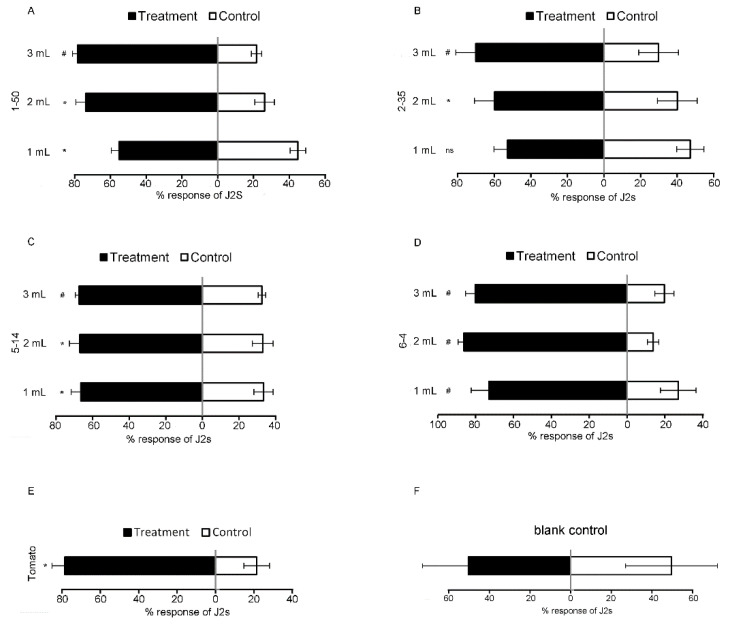
Responses of *M. incognita* to 4 strains in pot assay I. Response of *M. incognita* infective juveniles (J2s) to different doses of (**A**) *Bacillus* sp. 1-50 compared to LB control; (**B**) *B. brevis* 2-35 compared to control; (**C**) *B. cereus* 5-14 compared to control; (**D**) *C. indologens* 6-4 compared to control; (**E**) tomato compared to control; (**F**) blank control (both sides are sterile soil). Each value represents the mean ± standard deviation (*n* ≥ 3), and the means were separated using *t*-test (* *p* < 0.05, ^#^
*p* < 0.0001, ns = not significant).

**Figure 4 microorganisms-11-02271-f004:**
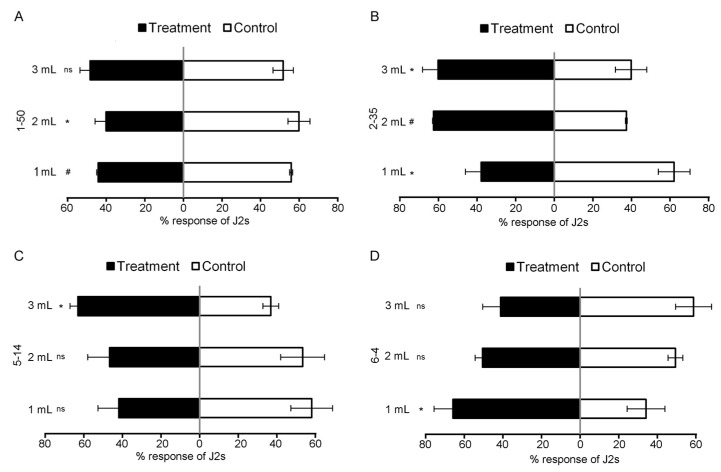
Responses of *M. incognita* to 4 strains in pot assay II. Response of *M. incognita* infective juveniles (J2s) to tomato with different doses of (**A**) *Bacillus* sp. 1-50 compared to control; (**B**) *B. brevis* 2-35 compared to control; (**C**) *B. cereus* 5-14 compared to control; (**D**) *C. indologens* 6-4 compared to control. Each value represents the mean ± standard deviation (*n* ≥ 3), and the means were separated using *t*-test (* *p* < 0.05, ^#^
*p* < 0.0001, ns = not significant).

**Figure 5 microorganisms-11-02271-f005:**
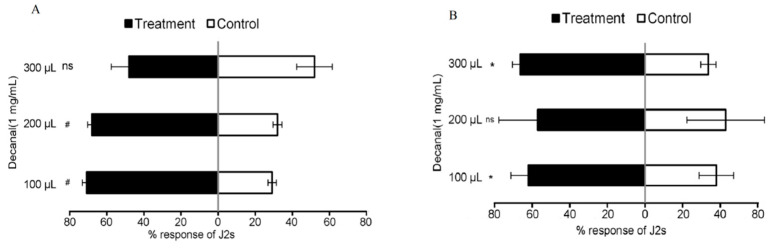
Responses of *M. incognita* to Decanal in pot assay I and assay II. (**A**) Decanal compared to control in assay I; (**B**) Decanal compared to control in assay II. Each value represents the mean ± standard deviation (*n* ≥ 3), and the means were separated using *t*-test (* *p* < 0.05, ^#^ *p* < 0.0001, ns = not significant).

**Table 1 microorganisms-11-02271-t001:** Identification of 4 isolates.

No.	Similar Strains	Similarity	NCBI Number
1-50	*Bacillus* sp.	98%	KP016607.1
2-35	*Brevibacillus brevis*	99%	AY882081.1
5-14	*Bacillus cereus* BS1	100%	KR063181.1
6-4	*Chryseobacterium indologens* McR-1	100%	JF894157.1

**Table 2 microorganisms-11-02271-t002:** Candidate VOCs produced via selected bacteria.

No.	Volatile Substance	Area Relative Content (%)/Retention Time (min)			
		1-50	2-35	5-14	6-4
1	Decanal	-	4.4/14.804	-	-
2	Benzeneacetaldehyde	1.75/8.025	1.13/8.225	-	-
3	Dimethyl disulfide	-	-	-	1.63/6.077
4	2-nonanone	-	-	-	14.1/11.007
5	Benzaldehyde	1.71/5.845	5.0/6.123	-	2.31/5.713
6	Indole	-	-	-	25.05/14.584
7	2,5-dimethylpyrazine	-	-	-	1.53/4.984
8	Pentadecanoic acid	4.16/25.056	-	5.42/25.095	-
9	Palmitic acid	7.92/27.353	-	3.05/27.295	-

-: This substance was not detected.

**Table 3 microorganisms-11-02271-t003:** Chemotaxis index of 9 VOCs candidates in different concentrations to *M. incognita*.

VOCs	Concentration (mg/mL)			
	1000	100	10	1
Decanal	0.00 ± 0.00	0.00 ± 0.00	0.01 ± 0.06	0.20 ± 0.10
Benzeneacetaldehyde	−0.15 ± 0.04	−0.10 ± 0.05	0.01 ± 0.02	0.07 ± 0.08
Dimethyl disulfide	0.14 ± 0.02	0.10 ± 0.02	0.03 ± 0.05	0.03 ± 0.11
2-nonanone	−0.01 ± 0.10	−0.03 ± 0.12	−0.02 ± 0.13	−0.07 ± 0.04
Benzaldehyde	−0.05 ± 0.12	−0.06 ± 0.16	−0.03 ± 0.12	−0.03 ± 0.10
Indole	-	−0.02 ± 0.06	0.03 ± 0.09	0.03 ± 0.10
2,5-dimethyl pyrazine	0.10 ± 0.01	0.06 ± 0.02	0.16 ± 0.10	−0.01 ± 0.01
Pentadecanoic acid	-	0.16 ± 0.02	0.06 ± 0.03	0.05 ± 0.02
Palmitic acid	-	0.11 ± 0.03	0.08 ± 0.06	0.07 ± 0.14

-: No data available.

## Data Availability

Data is unavailable due to privacy restrictions.

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
