# Peer review of "Chemotaxis of Meloidogyne incognita Response to Rhizosphere Bacteria"

_microorganisms, 2023, doi:10.3390/microorganisms11092271_

Round 1
Reviewer 1 Report
The research is very interesting and perspective to biocontrol technology. Especially important that the authors studied the effects not only for a single stimulus, but in combination with the plant host roots. The MS can be accepted after corrections. Major changes are to be done in sample sizes (n) for statistics.
P2L 66- 67
Each strain was calculated from three plates (counting nematodes under a microscope) and replicated three times. = Chemotaxis Assay in the Petri dishes. n=3 is too low sample for a statistical decision, in the pot experiments you use n=5, please explain.
= Please explain more thoroughly the experiment (Chemotaxis Assay in the Petri dish) design, the only reference to publication is not enough for readers.
P3 L 121
The soil on both sides was soaked and J2s were calculated by electron microscope.
= Why you use the electron microscope to count nematodes? What the method of counting?
P4 L136 (Fig. 1 legend)
= Error bars mark the standard deviation (SD) values? Please explain in the legend. Why the error bars are one-direction only, and not at both sides of the mean?
P5L149
M. incognita was highly attracted by decanal (1 mg/mL, C.I. = 0.20 ± 0.10) and trans-2-hexenal (10 mg/mL, C.I. = 0.23 ± 0.08).
= trans-2-hexenal is not mentioned as the selected and tested VOC and is absent in the Tables 2 and 3 and Fig 2
P5 L150
M. incognita was weakly attracted by pimethyl disulfide (100 mg/mL, C.I. = 0.14 ± 0.02), hexanal (100 mg/mL, C.I. = 0.12 ± 0.03),
= Hexanal is absent in the Tables 2 and 3 and Fig 2;
= pimethyl disulfide – is the dimethyl disulfide??
P5 L159-P6L160
The highest percent increases in attraction over the controls (P < 0.05) were 267 ± 0.61%, 172 ± 1.29%, 108 ± 0.20% and 545 ± 1.45% for Bacillus sp. 1-50, B. brevis 2-35, B. cereus 5-14, and C. indologens 6-4 respectively.
= Please explain how the values (in %) mentioned in the text correspond to the diagrams in Fig.3, where you use different concentrations (1,2,3 ml) of the treatment compounds; how you estimate the sample size (n) in the sentence above?
P7 L 186-190
Meanwhile, we also explored the chemotaxis effects of volatile compounds (decanal) on J2s in the presence or absence of tomato plants. The result of the decanal showed that the J2s preferred decanal compared to the control (P < 0.0001), with the highest percent increase of 145 ± 0.27% (Fig. 5A, B). In the preference for tomato, the J2s preferred the tomato with decanal relative to the control (P < 0.05), and the highest increase was 100 ± 0.34% (Fig. 5C).
= Please explain how the values (in %) mentioned in the text correspond to the diagrams in Fig.5, where you use different concentrations (100,200,300 µl) of the decanal treatment compounds; how you estimate the sample size (n) in the sentence above?
= Fig. 5C is absent, please correct the reference in Line 190 for Fig. 5B (assay II with the tomato combination);for line 189 the reference (Fig. 5A, B) is to be checked, while the talk is about Fig. 5A only (assay1, without tomato seedling roots)
Author Response
请参阅附件。

Reviewer 2 Report
See attached docs file

See attached file
Author Response
Please see the responses in the attachment
